

# The interdecadal worsening of weather conditions affecting aerosol pollution in the Beijing area in relation to climate warming

Xiaoye Zhang[1,2], Junting Zhong[1], Jizhi Wang[1], Yaqiang Wang[1], Yanju Liu[3]

[1] *State Key Laboratory of Severe Weather & Key Laboratory of Atmospheric Chemistry of CMA,*
*Chinese Academy of Meteorological Sciences, Beijing, 100081, China*
[2] *Center for Excellence in Regional Atmospheric Environment, IUE, CAS, Xiamen 361021, China*
[3] *National Climate Centre, China Meteorological Administration, Beijing, 100081, China*

## Abstract

The weather conditions affecting aerosol pollution in Beijing and its vicinity (BIV) in wintertime have worsened in recent years, particularly after 2010. The relation between interdecadal changes in weather conditions and climate warming is uncertain. Here, we analyze long-term variations of an integrated pollution-linked meteorological index (which is approximately and linearly related to aerosol pollution), the extent of changes in vertical temperature differences in the boundary layer (BL) in the BIV, and northerly surface winds from Lake Baikal during wintertime to evaluate the potential contribution of climate warming to changes in meteorological conditions directly related to aerosol pollution in this area; this is accomplished using NCEP reanalysis data, surface observations, and long-term vertical balloon sounding observations since 1960. The weather conditions affecting BIV aerosol pollution are found to have worsened since the 1960s as a whole. This worsening is more significant after 2010, with $PM_{2.5}$ reaching unprecedented high levels in many cities in China, particularly in the BIV. The decadal worsening of meteorological conditions in the BIV can partly be attributed to climate warming, which is defined by more warming in the higher layers of the boundary layer (BL) than the lower layers. This worsening can also be influenced by the accumulation of aerosol pollution, to a certain extent (particularly after 2010), because the increase in aerosol pollution from the ground leads to surface cooling by aerosol-radiation interactions, which facilitates temperature inversions, increases moisture accumulations, and results in the extra deterioration of meteorological conditions. If analyzed as a linear trend, weather conditions have worsened by ~4% each year from 2010 to 2017. Given such a deterioration rate, the worsening of weather conditions may lead to





corresponding amplitude increase in PM$_{2.5}$ in the BIV during wintertime in the next five years
(i.e., 2018 to 2022). More stringent emission reduction measures will need to be conducted by
the government.
## Introduction
Since individuals experienced heavy aerosol pollution episodes (HPE$_S$) in January 2013
in Beijing and its vicinity (BIV) in central-eastern China, changes in aerosol particle
concentrations and their chemical components have attracted special attention to high
population density areas with rapid economic growth (Huang et al., 2014;Zhang et al.,
2013;Guo et al., 2014;Wang et al., 2014a;Wang et al., 2014b;Wang et al., 2015;Sun et al.,
2014). However, these studies were mainly concerned with changes in emission sources and
changes in atmospheric physio-chemical characterizations. In addition, weather conditions
have an important impact on air pollution. Different weather conditions affect atmospheric
pollution by changing ventilation efficiency (i.e., winds, boundary layer height, convection, or
frontal passages), dry/wet deposition, loss ratios of chemical conversion, natural emissions,
background concentrations (Li et al., 2005;Liu et al., 2003;Leibensperger et al., 2008), early
morning solar radiation, frontal passage days (Ordonez et al., 2005), surface temperature and
relative humidity (Camalier et al., 2007). Specifically, aerosol pollution in Beijing has been
possibly affected by southerly/southwesterly surface winds . Aerosol pollution was also found
to become increasingly serious during recent decades (Zhang et al., 2015), which is partially
due to increasing emissions in air pollutants from anthropogenic activities (e.g., traffic,
industry, and power plants) (Li et al., 2017), but it is also influenced by regional and
unfavorable weather conditions (Zhang et al., 2015). Questions have been raised regarding
changes in weather conditions that affect HPEs in the BIV from a long-term perspective and
the effect of climate warming on meteorological factors that aggravate/alleviate aerosol
pollution in this area.
Here, we try to find a quantitative link between climate warming and unfavorable
weather conditions in the BIV from an interdecadal scale perspective by investigating
available surface and upper-air observations of different meteorological factors; this type of
study has not been conducted much so far. We use long-term balloon sounding observations,
particularly for temperature change in different layers, to analyze the vertical diffusion of
conditions and northerly winds from Lake Baikal (which is located in Beijing's cold air upper
transport pathway) to measure horizontal diffusion conditions. Since HPEs in the BIV usually
appear in winter, we focus our research on January data since 1960.
## Methods
**An index of meteorological conditions, PLAM (Parameter Linking Air Quality and**
**Meteorological Elements), which is almost linearly related with PM pollution, is used to**
**reveal changes in regional unfavorable weather conditions that affect aerosol pollution**



**in the BIV.** The formation and accumulation of aerosol pollutants are closely related with
various meteorological factors. However, a single factor cannot completely and linearly
reflect pollution conditions, and the effect of some factors even counteract or offset one
another (Sui et al., 2007;Pang et al., 2009). To describe meteorological conditions that change
simultaneously with identical amplitudes for PM mass concentrations during HPEs in winter
in the BIV, we use one comprehensive meteorological index (PLAM), which mainly indicates
regional atmospheric stability and air condensation ability, to reveal changes in regional
unfavorable weather conditions that affect heavy pollution in the BIV. The PLAM was
derived based on the relationship between PM mass concentrations and key meteorological
parameters from 2000 to 2007 for various regions in China (Che et al., 2009;Wang et al.,
2012;Wang et al., 2013).
It was established as a function of the following parameters:
$$\text{PLAM }(F)\ \in f(p, t, w, \text{rh}, e, s, c', ..)\qquad(1)$$
where $p$, $t$, $w$, rh, $e$, $s$, and $c'$ represent air pressure, air temperature, wind, relative
humidity, evaporability, stability, and the effective parameter associated with the contribution
of air pollution $\beta(c')$, respectively. Furthermore, the final PLAM can be attributed to two
major separate factors: (1) initial meteorological conditions $\alpha(m)$ associated with atmospheric
condensation processes and (2) a dynamic effective parameter associated with the initial
contribution of air pollution $\beta(c')$, which can be expressed as follows

$$\text{PLAM} = \alpha(m) \times \beta'(c).\qquad(2)$$

This index has been employed to evaluate the contribution of meteorological factors to
changes in atmospheric composition and optical properties over Beijing during the 2008
Olympic Games (Che et al., 2009), identify the contribution of specific meteorological factors
to a 10 d haze-fog event in 2013 (Zhang et al., 2013), estimate the relative contribution of
meteorological factors to changes in aerosol mass concentrations and chemical compositions
in different regions of China during winter from 2006 to 2013(Zhang et al., 2015) and
distinguish the feedback effect of meteorological conditions on the explosive increase in
$PM_{2.5}$ mass concentration during accumulation stages in the Beijing area (Zhong et al.,
2017;Zhang et al., 2017).
Because weather conditions that affect Beijing simultaneously affect a relative large area,
including Jing-Jin-Ji (i.e., Beijing, Tianjin and Hebei Province) and its adjacent areas
(including the Shandong, Henan Provinces and the Guanzhong Plain) in China (Zhang et al.,
2012), we use the PLAM determined by meteorological data from an observatory in Beijing



to represent regional unfavorable weather conditions, which are closely related to aerosol
pollution in the BIV.

3        HPEs often occur in wintertime; therefore, we compared the average PLAM in winter

with the other three seasons from 2013 to 2017 (Figure 1). It was found that adverse weather
conditions in winter are 1.4 to 2 times worse than those in other seasons, which indicates that
even if no additional pollution sources were added in winter (e.g., heating), $PM_{2.5}$ mass
concentrations are going to increase by at least 40% to 100% on average in winter simply
from unfavorable weather conditions. Here, we use the PLAM in January to explore changes
in meteorological conditions during HPEs in wintertime. Observations from the observatory
(54511) in southern Beijing for 57 years (from 1960 to 2017) were used to calculate the
PLAM and analyze its long-term changes

13       *Insert [Figure 1] here*

**Vertical temperature anomalies:** Atmospheric vertical observations at standard isobaric
surfaces were measured twice daily at 0800 Beijing time (BT) and 2000 BT; factors measured
included winds, temperature and relative humidity (RH) at the observatory (54511) in the
southern part of Beijing in January from 1960 to 2017. Based on the climatological mean
temperature in January, which was calculated as the 30-year atmospheric climate basic state
(i.e., 1960-1989), the temperature anomaly ($\delta T$) from 1960 to 2017 at different pressure layers
(1000 – 100 hPa) was calculated.

**Northerly winds from Lake Baikal:** Based on the NCEP/NCAR reanalysis data, we defined
the mean northerly wind velocity from Lake Baikal in January as an indicator for the effects
of winter monsoons on pollution-linked weather conditions in the BIV.
# Results and Discussion
**Weather conditions linked to aerosol pollution in the BIV in wintertime have worsened**
**since the 1960s, and the worsening is more obvious after the 1980s.**

Observed January PLAM values in Beijing exhibited an increasing trend from 1960 to 2017;
particularly, positive anomalies have occurred since the 1980s, which shows that weather
diffusion conditions favoring aerosol pollution in wintertime have strikingly worsened since
the 1980s (Figure 2a). Meanwhile, China's reform and opening up began nearly 40 years of
rapid economic growth, with a large amount of energy consumption with coals as the major
part. For example, in the year of 1980, China consumed approximately 0.6 billion tons of
coal. By 2013, China's total coal consumption was approximately 2.5 billion tons, which is a
4-factor increase (NBS-China, 2014). Because the PLAM primarily reflects the stability of air
masses and the condensation rate of water vapor on aerosol particles, it is linearly related to
the PM mass change (Wang et al., 2012;Che et al., 2009;Wang et al., 2013). Approximately
20% of increasing PLAM values since the 1980s, when calculated with a linear trend (Figure
2a), have been thought to cause an increase in $PM_{2.5}$ with similar amplitudes; this 20% change



has been considered to be only caused by intensive unfavorable weather. It is no wonder that
in the case for continued and increased emissions, when coupled with worsening weather
conditions, the upper limit of the environmental capacity in the BIV was exceeded in January
of 2013; Ten days of severe aerosol pollution first appeared in central-eastern China, with the
most serious pollution appearing in the BIV.
7       *Insert [Figure 2] here*
Based on the average interdecadal change in the PLAM during wintertime (Figure 2b), it can
be seen that the PLAM has been increasing since the 1960s. Particularly, in the last 8 years
between 2010 and 2017, the mean of PLAM increased larger than the growth rate of the mean
of the previous each ten years, which exhibited more noticeable unfavorable weather
conditions. When the $PM_{2.5}$ mass pollution accumulated to a certain extent, it caused the
further deterioration of weather conditions, which has been found in almost all HPEs in the
Beijing area since 2013 (Zhong et al., 2017;Zhang et al., 2017;Zhong et al., 2018). Therefore,
we hypothesized that the substantial rise in mean PLAM between 2010 and 2017 should have
benefited from the further worsening of meteorological conditions caused by higher $PM_{2.5}$
mass concentrations that reached a certain extent. In the BIV, aerosol pollution has become
increasingly serious during the past decades, particularly since 2010 (Zhang et al., 2015); in
January 2013, February 2014, December 2015, December 2016 to 10 January, 2017, 12
persistent HPEs occurred in Beijing, and the mass concentrations of $PM_{2.5}$ were high at
historically high levels (Zhong et al., 2018). There will be a detailed discussion on this issue
in a later section.
**The decadal worsening of meteorological conditions in the BIV was partly attributed to**
**climate warming**
Climate warming has a series of consequences. The vertical gradient of atmospheric
temperature decreases with the influence of climate warming (Dessler and Davis, 2013;Held
and Soden, 2006). The decadal warming is accompanied by increases in mid and upper
tropospheric specific humidity. The warmer the atmosphere is, the smaller the temperature
gradient is, and the more stable the atmosphere is, the greater the accumulation of air
pollution in the surface boundary layer. In this study, it can be seen that the relative upper BL
in Beijing is warmer than the lower layer (Figure 2c-d), which is indicative of the climate
warming phenomenon in the BIV. By analyzing 49 pollution episodes, Wu (2017) found that
the occurrence of pollution accumulation often caused by the occurrence of high-level
convergence layer in the context of climate warming. Weak westerly or northwesterly winds
dominate in the mid-upper troposphere and a convergence layer appears between 500 hPa and
700 hPa (Wu et al., 2017), which produce persistent and strong sinking motion in the mid-
lower troposphere to reduce the BL height and accumulate pollutants (Wu et al., 2017). As a
result of air masses sinking in the mid-lower troposphere, diverging in the lower layers, and
being warmed by adiabatic compression, a subsidence inversion appears in the lower layers,
which facilitates pollutant accumulation.


In Figure 2c, we found that the monthly mean temperature anomalies below 200 hPa
exhibited warming in some years since 1960, despite the inter-annual variability. The
difference in temperature anomalies between 1000 hPa and 850 hPa decreased throughout the
time period since 1960 when described by a linear trend (Figure 2d), which indicated that
temperature differences between the upper and lower boundary layers gradually declined in
the BIV, resulting in a more stable atmospheric stratification in this region. Because PLAM
anomalies gradually became positive after the 1980s (Figure 2a), temperature anomalies
between 1000 hPa and 850 hPa also became negative approximately after the 1980s (Figure
2d); this exhibit again that weather conditions after the 1980s, when China's reform and
opening up led to the formation of more aerosol pollution, worsened compared to those before
the 1980s within the context of climate warming. The correlation coefficient between the
monthly mean PLAM and the temperature anomalies difference between 1000 hPa and 850
hPa since the 1980s was -0.71 (Figure 3), which suggests that weather conditions most
directly related to pollution in Beijing (PLAM) were indeed closely related to climate
warming. With ~0.5 of the explained variance, one can believe that the contribution of
temperature differences due to climate warming to the continued increase in Beijing's PLAM
is around 50% in the month of January since the 1980s.

*Insert [Figure 3] here*

**The decadal worsening of meteorological conditions, especially when aerosol pollution**
**increased to a certain extent after 2010, may also be partly related to aerosol pollution,**
**which induces further worsening of meteorological conditions.**

The larger rise in the PLAM mean value between 2010 and 2017 in the BIV (Figure 2b) can
be considered to be partly attributed to a vicious cycle in meteorological conditions, which
resulted from aerosol pollution increasing to a certain extent. This can also be explained in
detail, as an example, in Figure 4.

*Insert [Figure 4] here*

We found surface cooling effect indicated by mean temperature anomalies, which was more
striking from 2010 to 2017 relative to that from 1980 to 2017. Aerosol pollution in the BIV
region has reached a very serious level since 2010 (Zhang et al., 2015), which was much
higher than that in the 1980s. Remarkably, more aerosols back-scattered a larger amount of
radiation into space, which caused a significant reduction in radiation reaching the ground.
This phenomenon (i.e., when aerosol pollution reaches a certain extent and results in a
temperature inversion near the surface layer, which causes more stable atmospheric
stratification) was widely found in a large number of HPEs in the BIV after 2013 (Zhong et
al., 2018).

A feedback loop of climate warming intensifying local unfavorable weather conditions,
forming aerosol pollution, and the accumulated aerosol pollution further exacerbating the
local unfavorable weather conditions and having vicious cycle of aerosol pollution is



illustrated in Figure 5. Climate warming via mid-upper tropospheric specific humidity
increasing and air adiabatic sinking to cause the upper atmosphere more warming relative to
lower one, easy to form unfavorable weather in the BIV to form aerosol pollution. During the
transport stage (TS) in pollution formed, relative strong southerly winds prevail in the lower
troposphere in the BIV, which transports pollutants and water vapor from the south of Beijing
to the urban area of Beijing. When the pollution accumulating to a certain extent during the
cumulative stage (CS), elevated PM2.5 established by the TS back-scatters amounts of solar
radiation to space due to its scattering property, which leads to near-ground radiative cooling.
This radiation reduction reduces near-ground temperature to facilitate anomalous inversion,
which subsequently suppresses vertical turbulent diffusion and decrease BL height to further
traps pollutants and water vapor. Induced by surface cooling, decreased saturation vapor
pressure substantially enhances RH. The joint effect of inversion suppression and surface
cooling results in appreciable near-ground moisture accumulation, which further accelerates
heterogeneous and liquid-phase reactions and enhance aerosol hygroscopic growth to increase
PM2.5 mass concentration. The noted positive meteorological feedback dominates PM2.5
explosive growth. (Zhong et al., 2017;Zhong et al., 2018;Zhang et al., 2017)
*Insert [Figure 5] here*
**The weakening of northerly wind affecting the BIV in wintertime also contributed to the**
**continuous deterioration of meteorological conditions in this area.**
Wind conditions represent one critical parameter in regulating the cycles of pollution episodes
in an area. Strong northerly winds and southerly winds closely correspond to clean periods
and pollution episodes in the BIV, respectively, because northerly winds (which originate
from less populated northern mountainous areas) carry unpolluted air masses, while southerly
winds carry polluted air masses from more populated and polluted southern industrial regions
(Jia et al., 2008;Guo et al., 2014;Zhong et al., 2018).
Because Lake Baikal is located in the upper transport pathway of northerly winds in winter
and is less affected by increasing/decreasing surface roughness in urban area, the northerly
winds from Lake Baikal substantially affect cold air mass movement to the North China Plain,
which further affects the formation and elimination of aerosol pollution in the BIV (Figure
2e). We found that monthly mean northerly wind speed from Lake Baikal has declined over
the past 57 years, particularly with respect to the past 27 years (i.e., since the 1980s). The
mean wind speeds during 1960-1969, 1970-1979, 1980-1989, 1990-1999, and 2010-2016 are
3.0 m s$^{-1}$, 0.92 m s$^{-1}$, 1.88 m s$^{-1}$, 2.11 m s$^{-1}$, 1.64 m s$^{-1}$, and 1.21 m s$^{-1}$ respectively (Figure 2f),
which indicate that the northerly wind speed has declined gradually as a whole since 1960. By
carrying less cold and dry air over the North China Plain, weakened northerly winds are
unfavorable for atmospheric diffusion. Over the past 37 years, the correlation coefficient
between northerly wind speed and PLAM is -0.63, which suggests that the year-to-year
variability of the northerly wind speed is closely associated with PLAM variability. The
number is statistically significant (p<0.1% for the correlation coefficient).



For changes in surface wind, MC Vicar (2012) found that a decrease in surface wind was
observed in major regions of the world (Mcvicar et al., 2012). Such surface wind trends can
be due to increasing surface roughness, the decrease in synoptic weather system intensity
and/or changes in mean circulation (Vautard et al., 2010). A variety of studies found that
surface winds decreased substantially in China (Xu et al., 2006;Guo et al., 2015;Chen et al.,
2013). In the urban area of Beijing, the decrease in winds below 300 m was considered to be
partly due to increasing surface roughness caused by land-use change(Liu et al., 2017).
However, this reason does not suffice when explaining wind speed changes from Baikal,
because the surface roughness of Lake Baikal has not been changed much due to less human
activities and industrial construction. The surface wind slowdown from Lake Baikal was
likely attributed to changes in atmospheric circulation, which can explain 10% to 50% of the
wind decline in the Northern Hemisphere (Vautard et al., 2010). In addition, the weakening of
the East Asian winter monsoon system (Niu et al., 2010) was responsible for the wind
slowdown. Both changes in mean circulation and decreases in winter monsoon system
intensity are consequences of climate warming.

## Summary

Changes in meteorological conditions in winter that are directly related to aerosol pollution in
the BIV have worsened since the 1960s. Particularly, positive anomalies have occurred since
the 1980s, which shows that weather diffusion conditions favoring aerosol pollution in
wintertime have strikingly worsened since the 1980s. Meanwhile, China's reform and opening
up began nearly 40 years of rapid economic growth, with large amounts of energy
consumption mainly deriving from coals. The decadal worsening of meteorological conditions
in the BIV was partly attributed to climate warming and may also partly be related to the
impact on aerosol pollution, which induces the further worsening of meteorological
conditions when increasing aerosol pollution to a certain extent (particularly after 2010). The
impacts of climate change on meteorological conditions that are directly related to aerosol
pollution in the BIV can also be verified in another aspect: the decrease in wind speed from
Lake Baikal in winter. Climate warming, characterized by an increase in warming in the upper
atmosphere compared to the low layer in the BIV, explained over 50% of the decadal
worsening of weather conditions that are directly related to aerosol pollution in the BIV; this
includes the part of weather condition worsening caused by the accumulation of aerosol
pollution to a certain extent. This worsening is unfavorable for the reduction of $PM_{2.5}$ mass
concentrations in the BIV in recent years; it even played a counter role, which probably led to
an approximate 4% increase in $PM_{2.5}$ mass concentration each year after 2010 when the linear
trend from 2010 to 2017 was taken into account. In the future, if the Chinese government
aims to maintain a decline in pollution, more effort is needed to offset the adverse effects of
climate warming.



**Acknowledgements:**
We thank for thoughtful discussion with Prof. Yihui Ding. This research is supported by the
National Key Project of MOST (2016YFC0203306), the Atmospheric Pollution Control of the
Prime Minister (DQGG0104), and the Basic Scientific Research Progress of the Chinese Academy
of Meteorological Sciences (2016Z001).





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

3

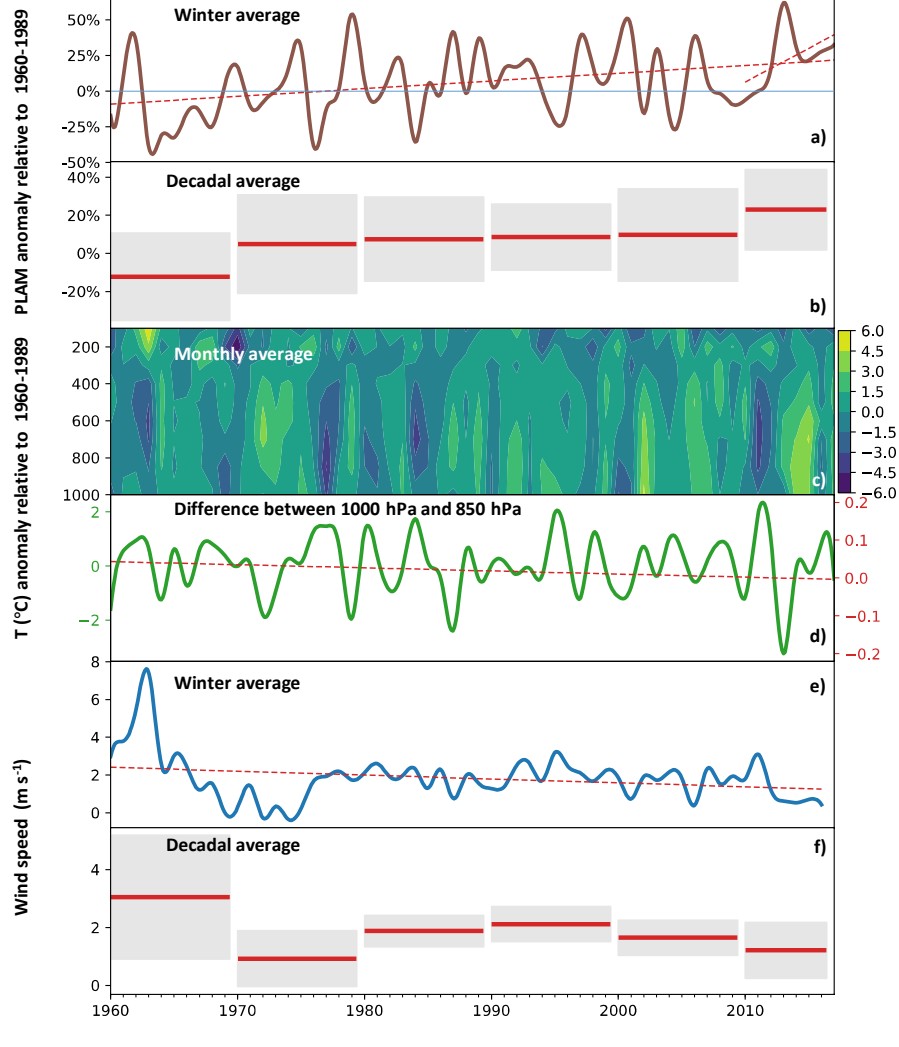





Figure 3. a): Correlations between the monthly mean PLAM anomalies and the temperature
anomalies difference between 1000 hPa and 850 hPa since the 1980s; b) correlations
between the monthly mean PLAM anomalies and wind speed from Lake Baikal since the
1980s

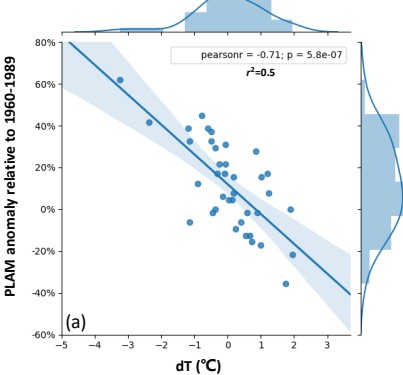
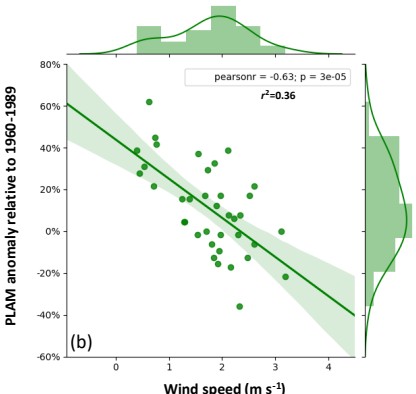



Figure 4. Temperature anomaly vertical profile. Blue denotes mean temperature anomaly from
1980 to 2017 relative to 1960 to 1989; Orange denotes mean temperature anomaly from
2010 to 2017 relative to 1960 to 1989)

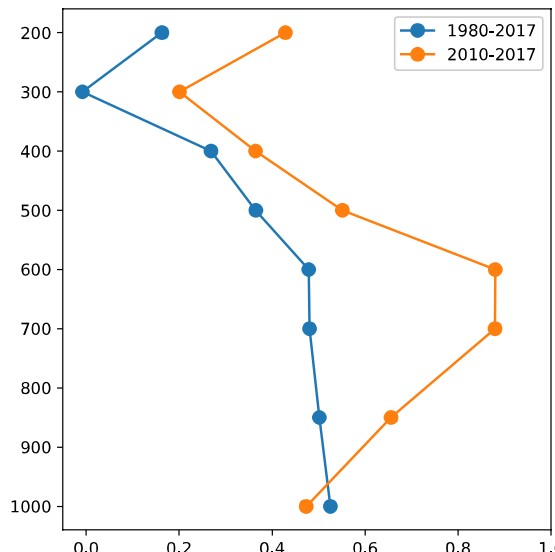





Figure 5. Schematic of important feedback loops for climate warming-unfavorable local and
regional weather conditions-forming and accumulating aerosol pollution-further
intensifying unfavorable weather conditions-more pollution.

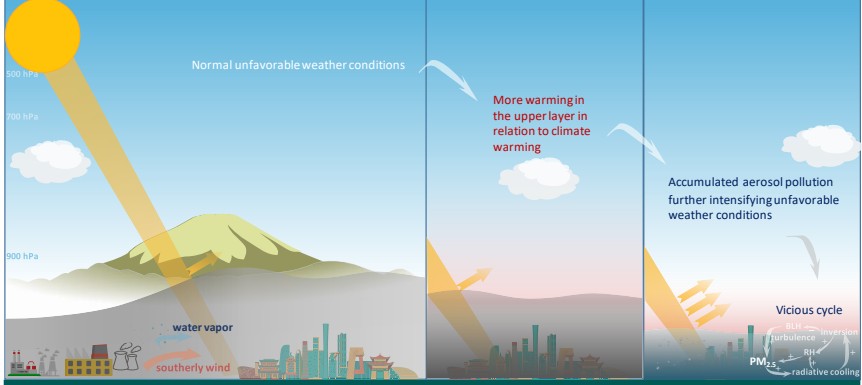

