# Peer review of "The interdecadal worsening of weather conditions"

_Atmospheric Chemistry and Physics, 2018_

## Referee Comment (RC1) · Anonymous Referee #1 · 27 Feb 2018

Comments to the Author (General comments) In this paper, the relationship between worsening of weather conditions and aerosol pollution, especially for the cause factor and feedback loop are investigated. I think it is a very interesting paper. There are some problems should clearly explain before this paper is accepted. 1.Page 2, in the introduction, the aerosol has direct indirect and semi-direct climate effect, especially for dust aerosols, which has been reviewed by Huang et al., (2014). The dust storm also has a significant effect on Beijing's air quality. The emission, transport and deposition of dust aerosols from Gobi Desert and Taklimakan Desert have been reported by previous papers (Chen et al., 2017; Huang et al., 2008). The references below are recommended. ïĄňHuang, J., T. Wang, W. Wang, Z. Li, and H. Yan (2014), Climate effects

of dust aerosols over East Asian arid and semiarid regions, J. Geophys. Res. Atmos., 119, 11,398–11,416, doi:10.1002/2014JD021796. ïĄňChen S., J. Huang, J. Li, R. Jia, N. Jiang, L. Kang, X. Ma, and T. Xie, 2017: Comparison of dust emissions, transport, and deposition between the Taklimakan Desert and Gobi Desert from 2007 to 2011. Science China Earth Sciences, doi: 10.1007/s11430-016-9051-0. Shao Y, Wyrwoll K H, Chappell A, Huang J, Lin Z, McTainsh G H, Mikami M, Tanaka T Y, Wang X, Yoon S. 2011. Dust cycle: An emerging core theme in Earth system science. Aeolian Res, 2: 181–204 2.Page 3, line 6-11, Could you explain what the high and low PLAM stand for. And if the high PLAM means the worse weather conditions and severe aerosol pollution. Since you said "The PLAM was derived based on the relationship between PM mass concentrations and key meteorological parameters . . .", I think the PLAM stands for the correlation between PM mass concentration and meteorological parameters and can't stands for the intensity of unfavorable weather conditions. There are some confusions, please explain clearly. 3.Page3, in equation (2), the detail calculation of the PLAM should be written. Where the $\alpha$(m) and $\beta$(c') come from and how to derive the two parameters? 4.In equation (2), the "$\beta$'(c)" should be changed to "$\beta$(c')". 5.In equation (1), the "," should be added before "(1)". 6.In line 16, "Where" should be the top lattice because this sentence is not ended. 7.Page4, line 1-2, why the PLAM can represent the unfavorable wether conditions, even aerosol pollution? 8.In Figure1, why you plot the first and fourth columns and what do them used for? 9.Page4, Line6-8, how can I derive these information from the figure 1? 10.I think the study areas should be illustrated in this paper and the location observation sites. 11.I think the areas of the northerly wind speed you calculated should be marked out. 12.Page 4, Line 27, the same to question 6. 13.Page 5, line 35, "Wu (2017)" should be corrected in to "Wu et al., (2017)". 14.In Fig.2a,d,e, if the linear trends pass the significant level ?

---

## Referee Comment (RC2) · Anonymous Referee #2 · 17 Mar 2018

The understanding of climate warming's impact on air quality is an important issue in atmospheric environment study. To explore this scientific issue, this paper investigated the relation of inter-decadal changes in climate warming and weather conditions for aerosol pollution in Beijing and the surrounding regions in North China Plain with frequent haze, presenting the interesting results, which could improve our understanding on climate and environment changes and fall within the scope of ACP. I suggest a few minor revisions before it is published as follows:

1) Please convert the format of manuscript according to the ACP manuscript introduction.

2) Please add the levels of significant test for the correlation coefficient and change trend analyses in the discussions.

---

## Author Comment (AC1) · 11 Apr 2018

Dear Anonymous Referee #1,

Thanks for your careful review of the manuscript. We read the comments carefully, and have responded and taken all of the comments into consideration and revised the manuscript accordingly. My detailed responses, including a point-by-point response to the review and a list of all relevant changes, are as follows:

"Anonymous Referee #1: (). In this paper, the relationship between worsening of weather conditions and aerosol pollution,

[Figure]

especially for the cause factor and feedback loop are investigated. I think it is a very interesting paper. There are some problems should clearly explain before this paper is accepted." "1. Page 2, in the introduction, the aerosol has direct indirect and semi-direct climate effect, especially for dust aerosols, which has been reviewed by Huang et al., (2014). The dust storm also has a significant effect on Beijing's air quality. The emission, transport and deposition of dust aerosols from Gobi Desert and Taklimakan Desert have been reported by previous papers (Chen et al., 2017; Huang et al., 2008). The references below are recommended. ï ËŽAnHuang, J., T. Wang, W. Wang, Z. Li, and H. Yan (2014), Climate effects of dust aerosols over East Asian arid and semiarid regions, J. Geophys. Res. Atmos., 119, 11,398–11,416, doi:10.1002/2014JD021796. ï ËŽAnChen S., J. Huang, J. Li, R. Jia, ËĞN. Jiang, L. Kang, X. Ma, and T. Xie, 2017: Comparison of dust emissions, transport, and deposition between the Taklimakan Desert and Gobi Desert from 2007 to 2011. Science China Earth Sciences, doi: 10.1007/s11430-016-9051-0. Shao Y, Wyrwoll K H, Chappell A, Huang J, Lin Z, McTainsh G H, Mikami M, Tanaka T Y, Wang X, Yoon S. 2011. Dust cycle: An emerging core theme in Earth system science. Aeolian Res, 2: 181–204"

Response: Yes, the effect of dust aerosols has been mentioned and supported by a certain number of literature (P2L50-53)

"2. Page 3, line 6-11, Could you explain what the high and low PLAM stand for. And if the high PLAM means the worse weather conditions and severe aerosol pollution. Since you said "The PLAM was derived based on the relationship between PM mass concentrations and key meteorological parameters . . .", I think the PLAM stands for the correlation between PM mass concentration and meteorological parameters and can't stands for the intensity of unfavorable weather conditions. There are some confusions, please explain clearly."

Response: PLAM is a weather index that is extracted from diagnostic analysis based on conventional meteorological elements. After its establishment, we tested it with PM mass concentration during 2000-2007, and found that there is a linear relationship

between PM and the PLAM. One of the core parameters of the PLAM is a function of water vapor condensation (it denotes the most important initial condition for secondary aerosol formation); another core parameter is the stability of regional air masses (it is the key element for formation of aerosol pollution in a region, and whether stratification is further stable after the formation of the pollution). Because the PLAM has a linear relationship with PM mass concentration. Therefore, the rising of PLAM indicates that meteorological conditions, which are most closely associated with aerosol pollution, is deteriorating. We added some explanations and the detail calculation of the PLAM in the text, to avoid confusion (P4L90-110). Actually, the PLAM has been widely used to reflect unfavorable weather conditions and to distinguish the change of PM mass caused by meteorological factors in the total change of PM (Zhang XY et al., 2009; Wang et al., 2012; 2013; Zhang XY et al., 2015; Zhong et al., 2017a; 2018a;b; Zhang Z et al., 2017)

"3. Page3, in equation (2), the detail calculation of the PLAM should be written. Where the $\alpha$(m) and $\beta$(c') come from and how to derive the two parameters? "

Response: Added (P4L90-110).

"4.In equation (2), the "$\beta$'(c)" should be changed to "$\beta$(c')".

Response: Changed.

"5.In equation (1), the "," should be added before "(1)".

Response: Revised.

"6.In line 16, "Where" should be the top lattice because this sentence is not ended. "

Response: Revised.

"7.Page 4, line 1-2, why the PLAM can represent the unfavorable weather conditions, even aerosol pollution?"

Response: The reasons for this have been given in the previous answer. One of the

core parameters of the PLAM is a function of water vapor condensation (it denotes the most important initial condition for secondary aerosol formation); another core parameter of PLAM is the stability of regional air masses (it is the key element for formation of aerosol pollution, and whether stratification is further stable after the formation of the pollution). PLAM index represents the most relevant meteorological element related to PM change. The rising of PLAM indicates that meteorological conditions, which are most closely associated with aerosol pollution, is deteriorating. Many research reports on the PLAM index that can reflect unfavorable weather conditions, which might be affected by aerosol pollution via the cooling effects of aerosols, have been published since the 2008 Beijing Olympic Games (Zhang XY et al., 2009; Wang et al., 2012; 2013; Zhang XY et al., 2015; Zhong et al., 2017a; 2018a;b; Zhang Z et al., 2017).

"8. In Figure 1, why you plot the first and fourth columns and what do them used for?"

Response: The purpose of plotting the first column is to compare with the second column. In the field of aerosol pollution research, people want to know whether the meteorological conditions of the late autumn (November) should be considered as the same as winter. After comparison, it is found that the meteorological conditions of the three seasons, which include the spring, summer and late autumn (Nov.) of the autumn, are basically the same as that of the included ones in autumn, and all are much better than the weather conditions in winter; The fourth column is the weather condition of the most polluted month since 2013 during winter, people in aerosol pollution research also want to know the difference in meteorological conditions between the most polluted month and whole winter season.

"9. Page4, Line6-8, how can I derive these information from the figure 1?"

Response: It is found that PLAM has a linear relationship with PM mass concentration through many comparisons with PM mass. So the range of PLAM changes can roughly represent the portion range of the change in PM comes from the contribution of unfavorable weather conditions.

[Figure]

"10.I think the study areas should be illustrated in this paper and the location observation sites."

Response: We added diagram (new Figure 1) gives the location of observation and the area of the north wind we calculate (P3L62-63; 66).

"11.I think the areas of the northerly wind speed you calculated should be marked out."

Response: We added diagram (New Figure 1) marked the area of the north wind we calculate (P3L62-63; 66).

"12.Page 4, Line 27, the same to question 6."

Response: Okay.

"13.Page 5, line 35, "Wu (2017)" should be corrected in to "Wu et al., (2017)"."

Response: Changed.

"14.In Fig.2a,d,e, if the linear trends pass the significant level?"

Response: Yes. We added some significant level description in the text.

---

## Author Comment (AC2) · 11 Apr 2018

Dear Anonymous Referees,

Thanks for your careful review of the manuscript. We read the comments carefully, and have responded and taken all of the comments into consideration and revised the manuscript accordingly. My detailed responses, including a point-by-point response to the review and a list of all relevant changes, are as follows:

"Anonymous Referee #2 The understanding of climate warming's impact on air quality is an important issue in atmospheric envi-

[Figure]

ronment study. To explore this scientific issue, this paper investigated the relation of inter-decadal changes in climate warming and weather conditions for aerosol pollution in Beijing and the surrounding regions in North China Plain with frequent haze, presenting the interesting results, which could improve our understanding on climate and environment changes and fall within the scope of ACP. I suggest a few minor revisions before it is published as follows:"

"1) Please convert the format of manuscript according to the ACP manuscript introduction."

Response: We have converted the format of this manuscript according to the ACP's standard format.

"2) Please add the levels of significant test for the correlation coefficient and change trend analyses in the discussions."

Response: Yes. We added some significant level description in the text.